# Three Types of Endometriosis: Pathogenesis, Diagnosis and Treatment. State of the Art

**DOI:** 10.3390/jcm12030994

**Published:** 2023-01-28

**Authors:** Ludovica Imperiale, Michelle Nisolle, Jean-Christophe Noël, Maxime Fastrez

**Affiliations:** 1OB GYN Departement, ULB—Université Libre de Bruxelles, H.U.B.—Hôpital Universitaire de Bruxelles, CUB Hôpital Erasme, Route de Lennik 808, 1070 Brussels, Belgium; 2Obstetrics and Gynecology Department, University of Liège, Boulevard du 12^ème^ de Ligne 1, 4000 Liege, Belgium; 3Pathology Department, ULB—Université Libre de Bruxelles, H.U.B.—Hôpital Universitaire de Bruxelles, CUB Hôpital Erasme, Route de Lennik 808, 1070 Brussels, Belgium

**Keywords:** endometriosis, peritoneal endometriosis, deep infiltrating endometriosis, ovarian endometriosis

## Abstract

At present, there is no curative treatment for endometriosis. Medical management and surgical treatment do not provide long-term relief. A detailed understanding of its pathophysiology is mandatory in order to facilitate both the diagnosis and treatment. The delay that typically precedes proper diagnosis (6 to 7 years) is probably one of the most challenging aspects of endometriosis management. In 2012, the total cost per woman due to endometriosis was estimated to be 9579€ per year in a multicenter study across the USA and Europe. According to their physiopathology and their localization, ectopic endometrial lesions, consisting of endometrial glands and stroma, can be divided into three different types: superficial peritoneal endometriosis (SPE), ovarian endometrioma (OMA), and deep infiltrating endometriosis (DIE). The following paper aims to review the available data in the literature on the pathogenesis, diagnosis, and treatment of different types of endometriosis.

## 1. Introduction

Endometriosis is a chronic inflammatory disease characterized by the presence of endometrium-like tissue outside the uterus [1]. This definition, however, is extremely reductive. Due to its complexity, the variety of its clinical presentations, and the still unclear physio-pathobiological origin, endometriosis represents, nowadays, a big challenge in the field of benign gynecological disorders. Despite its first appearance in the scientific literature in 1860, defined and described by Karl von Rokitansky, not much progress has been made in the knowledge of its pathophysiology, diagnosis, and treatment [2].

The diagnosis is usually made through surgical visualization, ideally laparoscopy, although the recent guidelines from ESHRE suggest that actual imaging modalities for some types of endometriosis could be sufficient in the diagnosis process [3].

At present, there is no curative treatment for endometriosis, and clinical management of symptoms such as pain occurs through medical and/or surgical treatments [4]. Medical treatment follows the basic principle of suppression of menstrual cycles, resulting in the reduction in inflammation and blocking the effect of oestrogens [Falcone 2018]. Unfortunately, current medical therapies could alleviate symptoms, but they are far from curative treatments and can often lead to side effects that compromise patient compliance [5]. Surgical treatment aims to remove bulky endometriotic lesions or, in more complex cases, complete excision of the pelvic organs [6]. Neither the medical management nor the surgery turns out to give relief in the long term or is universally acceptable for the patients [7].

The following paper aims to review the available data in the literature on the pathogenesis, diagnosis, and treatment of the different types of endometriosis.

In the present scientific literature, there is no review that systematically describes the three types of endometriosis. The importance of this article lies in the fact that through a review of the literature, it deals in a didactical way with the pathogenesis, diagnosis, and treatment of the three types of endometriosis.

## 2. Materials and Methods

### 2.1. Search Strategy

We searched up to July 2022 the PubMed database (National Library of Medicine, https://pub-med.ncbi.nlm.nih.gov/ accessed on 31 July 2022). A combination of Medical Subject Headings (MeSH) descriptors has been used: “endometriosis”, “peritoneal endometriosis”, “three types of endometriosis”, “ovarian endometriosis”, “deep infiltrating endometriosis”, “pathogenesis of endometriosis”, “diagnosis of endometriosis”, “treatment of endometriosis”.

In addition, we manually searched references for other relevant publications.

### 2.2. Screening of Publications

After the systematic search using the specific MeSH, the authors performed a global screening based on the published literature, including publications, and articles recognized as relevant, which were analyzed in more detail. All prospective, retrospective studies and reviews of the literature published in English were included.

A total of 375 articles were identified upon initial search. Following exclusions and a search of additional relevant publications, 39 eligible articles were identified, 32 studies, and 17 reviews (Figure 1).

## 3. Different Types of Endometriosis

According to their physiopathology and their localization, ectopic endometrial lesions, consisting of endometrial glands and stroma, can be divided into three different types: superficial peritoneal endometriosis (SPE), ovarian endometrioma (OMA) and deep infiltrating endometriosis (DIE) [8].

### 3.1. Peritoneal Endometriosis

Peritoneal endometriosis can be found in 15–50% of all women diagnosed with endometriosis [9].

#### 3.1.1. Pathogenesis

Nisolle et al. classified the peritoneal lesions into three different kinds: red, black, and white. They represent the evolutionary steps of the disease. Red lesions represent the first step of peritoneal endometriosis and are highly vascularized and active lesions [8].

Black lesions represent the second step, advanced endometriosis, and finally, white lesions are quiescent endometriosis or healed endometriosis, or latent lesions [8].

How do these endometriotic cells arrive in the peritoneum? Retrograde menstruation is probably the pathogenic hypothesis supported by the strongest evidence. Endometrial fragments driven by uterine contractions through the fallopian tubes reach the peritoneal cavity and implant themselves [10].

Retrograde menstruation among women is quite a common event [11]. Other factors, therefore, probably permit endometrial cells to attach to peritoneal surfaces and become endometriotic lesions. Another concomitant mechanism could be the transformation of the peritoneal mesothelium into glandular endometrium (celomic metaplasia) [12]. Since endometrial stem-cell and progenitor-cell populations were found in eutopic endometrium [13], we can argue that these cells, helped by retrograde menstruation, may be involved in the development of peritoneal endometriosis [14]. Among women with endometriosis, endometrial stromal cells present an altered integrin profile which gives them an inherent adhesive capacity [15], which is amplified by the inflammatory environment typical of endometriosis. The proliferation of endometriotic tissue is promoted by estradiol. Endometriotic lesions show an increased expression of aromatase, estrogen receptor β, and steroidogenic acute regulatory protein [16] and decreased expression of 17β-hydroxysteroid dehydrogenase 2 [17]. Additionally, ectopic endometrial cells are characterized by a “progesterone resistance”, which consists of epigenetic differential methylation of PR-B, HOX, and GATA family transcription-factor genes and a consequently altered progesterone signaling [18], dysregulating endometrial decidualization and inhibiting estrogen-dependent epithelial-cell growth [19]. All of these processes are supported and amplified by a localized inflammatory and immune reaction through the overproduction of prostaglandins, chemokines, and cytokines [19].

#### 3.1.2. Diagnosis

Unfortunately, we still need to perform a laparoscopy to detect peritoneal endometriotic lesions. To date, an imaging method does not exist that can be compared, for sensitivity and specificity, to laparoscopy [20]. Transvaginal ultrasound resulted in poor sensitivity (65%; 95%CI 27% to 100%) but good specificity (95%; 95%CI 89 to 100%). MRI resulted in low specificity and low sensitivity (72% and 79%, respectively) in diagnosing peritoneal endometriosis [3].

Two small studies conducted by Manganaro in 2012 and by Thomer in 2014, using 3.0 tesla MRI, reported sensitivity between 81–95% and a specificity of 100% [21,22]

#### 3.1.3. Treatment

To date, there are no studies specifically exploring the effect of surgery or medical treatment among women suffering only from SPE. In the literature, some studies included only women with ASRM (American Society for Reproductive Medicine) stage I and II endometriosis, and most of these may have SPE. Nevertheless, it would be very difficult to generalize the results of these studies to women with SPE only since ASRM stage I and II disease may include ovarian endometriomas (<1 cm) or deep infiltrating endometriosis [3].

Several national and international guidelines recommend laparoscopic treatment of minimal or mild endometriosis in women experiencing infertility and/or pain associated with endometriosis [23,24]. Nonetheless, only one RCT found differences in overall pain using daily pain measurements at different time points [25].

Lately, progress has been made in the field of non-hormonal medical treatment of endometriosis. New therapeutic pathways, such as the use of antiangiogenic drugs, could be interesting for treating SPE.

Antiangiogenic agents, such as soluble fms-like tyrosine kinase 1 (sFlt-1), antibodies targeting VEGF (two VEGF-A antagonists), and galectins (Gals—glycan-binding proteins that bind specifically to β-galactosides), in mouse models, were shown to be able to reduce endometriotic lesions. These molecules are very active on hypervascularized lesions but not fibrotic lesions and adhesions [26]. Due to these characteristics, it could be interesting to study them in women suffering from peritoneal endometriosis exclusively or as first-line drugs to prevent deep and ovarian endometriosis.

### 3.2. Ovarian Endometrioma:

Ovarian endometrioma can be found in 2–10% of women of childbearing age and 50% of women treated for infertility [10].

#### 3.2.1. Pathogenesis

Ovarian endometrioma (OMA) is defined as ovarian cysts covered by an endometrial epithelium containing thick, brown fluid [8]. Different pathogenetic hypotheses, based on the histological observations of the cyst wall and the cyst internal cover, have been hypothesized to explain the origin of OMA.

##### Superficial Implants and Invagination of the Ovarian Cortex:

Brosens suggest that adhesions between the ovary and the peritoneum of the posterior leaf of the broad ligament would facilitate the development of endometriotic implants deriving from retrograde menstruation. Then, the ovarian cortex would invaginate, and the accumulation of menstrual debris from bleeding of the active implant invading the surrounding ovarian cortex would produce the typical “chocolate” liquid [27].

##### Superficial Implant and Corpus Luteum Invasion:

An alternative to the previous theory proposed by Vercellini and co-workers assumes that the entrapped blood originated from a corpus luteum that does not undergo resorption due to the presence of endometriotic lesions and adhesions on the ovarian cortex [28].

##### The Metaplasia of the Invaginated Mesothelial Inclusions

Donnez et al. described the presence of ovarian epithelial invaginations in a continuum with ectopic endometrial tissue. A possible explanation, according to these authors, could be the celomic metaplasia of the mesothelium covering the ovary that invaginates into the ovarian cortex forming mesothelial inclusions [29].

#### 3.2.2. Diagnosis

Nisenblat et al. conducted a Cochrane review which showed for a transvaginal ultrasound a good sensitivity and specificity with acceptable heterogeneity and confidence interval (sensitivity 93%, 95%CI 87 to 99%; specificity 96%, 95%CI 92 to 99%;) [30]. The results for MRI were similar to those from transvaginal ultrasound (specificity 91% and sensitivity 95%) in diagnosing OMA. Bazot et al. compared transvaginal and transrectal ultrasound with MRI. Transrectal ultrasound was less sensitive and less specific (89% and 77%, respectively) compared to transvaginal ultrasound (94% and 86%, respectively) and MRI (92% and 88%, respectively), which instead were both promising [31].

Based on these studies, we can state that transvaginal ultrasound and MRI are two valid diagnostic tools for detecting ovarian endometrioma.

#### 3.2.3. Treatment

To date, three different laparoscopic surgical approaches are proposed in the literature for the surgical treatment of ovarian endometrioma: surgical excision (cystectomy), drainage and coagulation of the cystic wall, and drainage followed by CO_2_ laser vaporization of the cystic wall [32].

Regarding endometriosis-associated pain recurrence and OMA recurrence, cystectomy is likely superior to drainage and coagulation in women with ovarian endometrioma (≥3 cm [32]. Compared to CO_2_ laser vaporization, cystectomy showed a similar recurrence rate [33] Surgery of endometriomas may have an important impact on future fertility. After bilateral ovarian endometrioma removal, the risk of ovarian failure is described to be 2.4% [34].

A paper recently published by Candiani et al. studied the reproductive outcome in infertile women with endometriomas managed by either CO_2_ laser vaporization or cystectomy. This prospective study on 142 patients with a history of surgical treatment of OMA revealed comparable chances of postoperative pregnancy in the two groups of patients (cystectomy versus CO_2_ laser vaporization). Women treated with CO_2_ laser vaporization had higher postoperative AMH (anti-mullerian hormone) and AFC (antral follicular count) levels than those who underwent surgical cyst removal and responded nearly one and a half times better to ART (assisted reproductive techniques) than those who underwent cystectomy. On the other hand, they were 30% less likely to conceive spontaneously [35].

None of the above-mentioned surgical techniques has shown to be superior in preserving fertility in women treated for ovarian endometriomas. Nevertheless, surgery for recurrent endometriomas is more damaging to healthy ovarian tissue and ovarian reserve than primary surgery [36]. A recent systematic review of the literature and meta-analysis reported that cystectomy for bilateral endometriomas has a detrimental and prolonged effect on ovarian reserve [37].

In order to preserve the ovarian reserve, nonsurgical treatment of ovarian endometriomas, such as aspiration or sclerotherapy, has been studied. The sclerotherapy technique was proposed for the first time by Okagaki in 1999 [38]. This technique consists of injecting a sclerosing agent into the cyst cavity. The disruption of the cyst epithelial lining, with subsequent inflammation and fibrosis, will result in the obliteration of the cyst [38]. Although it has been demonstrated effective and cost-effective [39], sclerotherapy has not been widely used.

In 2021, Alborzi et al., in a prospective cross-sectional study, showed equal results in ART outcomes comparing sclerotherapy to laparoscopic cystectomy. However, the recurrence of the disease in the sclerotherapy group was significantly higher [40].

We can argue that for patients who need to go on ART, sclerotherapy should be an interesting option.

### 3.3. Deep Infiltrating Endometriosis:

The rate of DIE among all patients affected by endometriosis is estimated at 20% resulting in a prevalence of 2% [41].

#### 3.3.1. Pathogenesis

Deep infiltrating endometriosis (DIE) is characterized by a specifical histologic pattern that includes well-differentiated glandular cells, pure stromal cells, glandular or mixed differentiated cells, and pure undifferentiated glandular cells [42]. Thus, undifferentiated endometriotic lesions are described as resulting from tissue resistance to the suppressive effects of peritoneal fluid, allowing these endometrial foci to infiltrate more deeply.

Zanatta et al. observed that, in rectosigmoid DIE lesions, the expression of ERa, PR-A, and PR-B mRNA is increased, suggesting a different behavior compared to ovarian and peritoneal endometriosis [43].

A hostile microenvironment, such as the vagina or the bowel, requires adaptation and immunotolerance for endometrial stroma and epithelial cells to survive and proliferate. A crucial role in these adaptations is probably played by Sphingosine-1-phosphate (S1P) receptor 1, which controls a very complex inflammatory pathway leading to the production of nitric oxide and prostaglandin, induced by interleukin (IL) 1b and tumor necrosis factor-a (TNF-a) [44]. In DIE, in particular, in the case of the most severe form with intestinal involvement, a study conducted in 2014 showed increased serum levels of IL-1b and IL-1 receptor type II (IL-1sRII) in patients with DIE [45].

The over-expression of vascular endothelial growth factor (VEGF) is a characteristic of endometriotic cells, and this is increased in red vascular peritoneal endometriotic lesions compared to older black or white scarred lesions. This expression differs between SPE, OMA, and DIE. Deep endometriotic lesions of the rectum, particularly, have been shown to have a higher expression of VEGF-A and VEGF receptor 2 and increased blood vessel density compared to ovarian or bladder endometriosis [11].

Deep endometriosis lesions are, therefore, the result of different pathogenic pathways, such as differentiation of undifferentiated cells, inflammation, and neovascularization. All these processes probably, allow the new endometriotic tissue to adapt better and to proliferate in inhospitable anatomical sites.

#### 3.3.2. Diagnosis

The peculiar difficulty of the diagnosis of deep endometriosis lies in the fact that it often has the characteristic of attacking pelvic organs, the abdominal wall, but also retroperitoneal structures [44].

Transvaginal ultrasound, including conventional ultrasound, 3D ultrasound, and sonovaginography, despite its limitation as an operator-dependant tool, in a study by Guerriero et al., showed a good specificity of up to 94% and a sensitivity of 79%, which may be hardly improved with 3D ultrasound (87%) [46].

A recent systematic review of the literature reported that there was no significant difference between MRI and transvaginal ultrasound in detecting recto-sigmoid endometriotic lesions [47].

#### 3.3.3. Treatment

Deep infiltrating endometriosis may affect several pelvic organs such as uterosacral ligaments, vagina, rectovaginal septum, pelvic side walls, ureter, bladder, or bowel. When surgical treatment is required, and excision of these lesions is needed, the surgeon is faced with a difficult challenge mainly because of the heterogenicity of the localization of the lesions. That leads to the difficulty of reproducing the same type of surgical technique each time.

It has been reported that the 5–12% of women affected by endometriosis have deep endometriosis involving the bowel [48]. When bowel infiltration is present, about 90% of the lesions are localized on the sigmoid colon or on the rectum. Segmental resection, superficial shaving, and discoid resection of the bowel are the current possible approaches for colorectal endometriosis nodules removal. To date, none of these surgical techniques has been shown to be superior in terms of recurrence rate, symptom relief, and quality of life improvement.

Bokor et al., in 2020, in a retrospective multicentric study, showed that, in patients undergoing rectal surgery for low DIE, LARS (low anterior resection syndrome) is less frequent after NVSSR (nerve- and vessel-sparing segmental resection) compared with a more conservative approach such as LTADE (laparoscopic-transanal disk excision) [49].

A randomized trial conducted by Roman et al. [50] showed higher rates of bowel stenosis requiring complementary procedures under general anesthesia for patients undergoing colorectal resection. They also found a risk of abnormal bowel movements in 40% of cases regardless of surgical management.

A recently published systematic literature review studied the impact of surgery on a very specific cohort: patients undergoing ART with infertility and DIE. The results were very consistent for all endpoints examined and showed a statistically significant benefit for surgery before ART, although they should be confirmed by RCTs [51].

Data also show that surgery improves the quality of life and pain in women with deep endometriosis, but the literature on these subjects lacks consistency. As surgery in this population of women is possibly associated with significant intraoperative and postoperative complication rates, the ESHRE guidelines recommend that this kind of surgery is performed in the center of expertise only after the patient has been informed of potential risks, benefits, and long-term effects, with a minimally invasive approach in a multidisciplinary setting with the goal to radically remove of all the endometriosis lesions [3].

In addition to significant pain relief, radical treatment of deep endometriosis also has a positive impact on fertility outcomes [52].

## 4. Conclusions and Future Perspectives

In spite of being a benign disorder, endometriosis is a difficult disease to diagnose and treat. A detailed understanding of its pathophysiology is urgently needed in order to facilitate both the diagnosis and treatment.

Endometriosis is an estrogen-dependent inflammatory disease. The ectopic survival of these apparently normal endometrial cells may result from an overreaction during endometrial regeneration in the uterus. Circulating endometrial progenitor and stem cells are enriched during the proliferative phase of endometrial regeneration. These cells may have a high ability to survive in an extrauterine site. In addition, local inflammatory pathways and immune surveillance normally function to erase these ectopic lesions, and the result of this combination of events determines the future of endometriotic lesions, which may be silent fibrotic lesions or active lesions.

Interestingly, recent advances in the molecular genetic analysis of clonal pathways in endometriosis development can lead us to a revision of our previous model of endometriosis pathogenesis, in particular regarding DIE lesions [53].

Regarding the diagnosis of endometriosis, the delay that typically precedes proper diagnosis (6 to 7 years) [54] is probably one of the most challenging aspects of managing this disease. There is a need for non-invasive diagnostic tools. Transvaginal ultrasound and magnetic resonance imaging have good sensibility and specificity in diagnosing OMA and DIE, but the non-invasive diagnosis of SPE remains a challenge. A possible area of investigation to follow to allow a non-invasive and more targeted diagnosis could be the combination of different non-invasive techniques such as PET (positron emission tomography) and MRI using specific tracers of endometrial tissue. For example, isotopic imaging of estrogen receptors has been reported in patients with breast cancer [55], and one study in 2014 used 16a-[18F]fluoro-17b-estradiol [18F]FES as a tracer as part of a diagnosis of endometriosis [56]. Another area of recent interest is the use of biomarkers. Studies have been conducted on genome-wide association. Understanding the role of single-nucleotide polymorphisms could lead to identifying a panel of biomarkers for a more accurate diagnosis with bigger specificity [57]. Improvement in all of these research areas is needed in order to accelerate diagnosis and also create opportunities for tailored treatments. The medical therapies available are mainly aimed at relieving pain. Unfortunately, 11% to 19% of women report no improvement in pain [58].

Probably when choosing medical treatments for pain associated with endometriosis, health professionals should not only take into account the effectiveness of the treatment but also side effects, tolerance and treatment compliance, costs, and preferences of women. If we consider the chronic nature of this disease, this seems very important and potentially determines a consistent improvement in the patient’s overall quality of life, sexual activity, social activities, mental health, and work.

Outcomes of surgical therapies are not better since 20% of women show no improvement, and the recurrence rates range from 30% to 50% [59]. Endometriosis surgery usually involves young women with the desire for a future pregnancy. The surgery could improve the quality of life and could reduce the intensity of painful symptoms. However, this beneficial effect may be negated if the surgery seriously alters the chances of conception irreversibly. Surgery should avoid damaging reproductive organs and should be carried out with the aim of preserving fertility and allowing the patients to achieve excellent results with spontaneous conception or by ART.

Exploring future treatment modalities that reduce infertility and pain is urgently needed.

With this review, we want to underline the complexity of this disease. It is probably necessary to start looking at endometriosis as a set of different sub-pathologies that can exist together or separately in the same patient and which probably do not have the same pathogenesis. Even if similar, they are not detected with the same diagnostic tools, and they need to be treated differently depending on their own peculiar characteristics.

Finally, it is primordial and now more than current to recognize that endometriosis has significant social, public health, and economic implications. It is probably time that healthcare systems give proper attention to this condition in order to recognize it as other chronic major diseases.

## 5. Limitations

The limitations of this article are represented by the fact that the review is a narrative review and not a systematic review with meta-analysis. Therefore, it lacks statistical and analytical value. Despite this, the present article has a descriptive value, and it has an up-to-date point of the current situation of the literature regarding the pathogenesis, diagnosis, and treatment of three types of endometriosis.

## Figures and Tables

**Figure 1 jcm-12-00994-f001:**
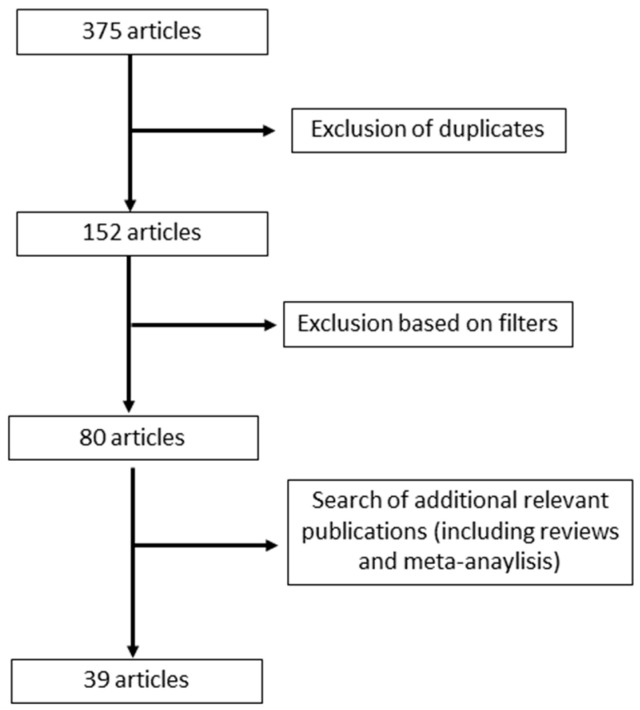
Flow diagram for the selection of the articles included in this review.

## Data Availability

Not applicable.

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
