# Peer review of "Three Types of Endometriosis: Pathogenesis, Diagnosis and Treatment. State of the Art"

_jcm, 2023, doi:10.3390/jcm12030994_

Round 1

Reviewer 1 Report

1. Some sentences have multiple references for example in page 2, line 56 and page 3, line 110 and ....

Please reform this multiple references and put proper references at the end of each sentence (please exert this note all over the manuscript)

2.Some sentences of the manuscript doesn't have appropriate reference. For exame in page 1, line 29-33, the sentence " Currently, there is no curative ... the effect of oestrogens"

Please add proper reference at the end of each sentence all over the manuscript (exept sentences that are well-established or the results of present panuscript)

3.In page 1, line 11-13, the authors have written about the problems in the treatments of endometriosis but the abstract of the manuscript should introduce a total concept to readers so that they can comprehend the totality of the manuscript. Your manuscript is about "Three types of endometriosis: pathogenesis, diagnosis and treatment." thus, please rewrite the part" abstract" based on the main idea of your manuscript (not just the treatment of endometriosis)

4.In page 1, line 11-13, the sentence " Neither the medical management nor the surgery turns out to give relief in the long term or is universally acceptable for the patients." is repeated again in line 34-36. Please reform this sentence in order to avoid self-plagiafism

5.in page 1, the part :Introduction" (line 19-42); the authors have written about endometriosis but this part does not have consistensy and continuity. Please rewrite this part of the manuscript according to order below:

First: write about the definition of endometrisis and challenges upon it's definition

Second: write about current various curative approaches in order to cure endometriosis and their weakpoints 

Third: write about the importance of the present manuscript based on previous information (in thr part Introduction) 

Note: authors in the part "introduction" should mention that why they have written this manuscript (the importance of this manuscript) 

6.in page 1, line 37-40 This part of introducrion disrupts the continuity of this part of the manuscript and does not have a proper connection with it's previous paragraph.Please rewrite or omit this part.

7.In page 2, line 57-60 Why authors have written this statistics in the part " 3. Different types of endometriosis"? It is better that this epidemiological data are written in their proper site of the manuscript.

8. In page 3, line 124-127 Please add suitable reference at the end of the sentences of this part.

9.Page 3, line 128, 135 and 140 Please write these three sentences in the form of subheading like:

3.2.1.1.Superficial implants and invagination of the ovarian cortex 

3.2.1.2.Superficial implant and corpus luteum invasion 

3.2.1.3.The metaplasia of the invaginated mesothelial inclusions

10. In page 4, line 145, part " 3.2.2. Diagnosis"

Please write the conclusion of this part based on scientific studies you have mentioned in this part.

11. Please add appropriate reference at the end of the sentences in page 4, line 156-159

12.In page 4, line 171, there are two abbreviations called "AFC" and "AMH"

please expand these two abbreviation in brackets.

13. In page 4, line 194, part "3.3.1. Pathogenesis"

Please write a conclusion for this part based on previous information of mentioned part. (Please write what you conclude of the process of pathogenesis of DIE based on the data you have written in this part)

14. In page 5, line 219-221, line 230-235 and line 237-241

Please add appropriate reference at the end of the sentence(s).

15. In page 6, line 275-277

Please omit or tranfer this part to it's proper 

section of manuscript. 

In the part "Conclusion" authors should talk about the results of their work. The comparison of the results with other studies belongs to previous sections of manuscript.

16.Please check references carefully (especially their titles and doi)

Author Response

Point 1: Some sentences have multiple references for example in page 2, line 56 and page 3, line 110 and Please reform this multiple references and put proper references at the end of each sentence (please exert this note all over the manuscript)

Response 1: Multiples references were reformed and proper references were put at the end of each senteces all over the manuscript

Point 2: Some sentences of the manuscript doesn't have appropriate reference. For example in page 1, line 29-33, the sentence " Currently, there is no curative ... the effect of oestrogens"

Please add proper reference at the end of each sentence all over the manuscript (except sentences that are well-established or the results of present manuscript)

 Response 2: Proper references were added at the end of each sentence all over the manuscript

Point 3: In page 1, line 11-13, the authors have written about the problems in the treatments of endometriosis but the abstract of the manuscript should introduce a total concept to readers so that they can comprehend the totality of the manuscript. Your manuscript is about "Three types of endometriosis: pathogenesis, diagnosis and treatment." thus, please rewrite the part" abstract" based on the main idea of your manuscript (not just the treatment of endometriosis)

 Response 3: Abstract was rewritten according to revision of reviewer 1 and 2 (cfr. Reviewer 2 point 1)

 Point 4: In page 1, line 11-13, the sentence " Neither the medical management nor the surgery turns out to give relief in the long term or is universally acceptable for the patients." is repeated again in line 34-36. Please reform this sentence in order to avoid self-plagiafism

 Response 4: The sentence was reformed

Point 5: in page 1, the part :Introduction" (line 19-42); the authors have written about endometriosis but this part does not have consistensy and continuity. Please rewrite this part of the manuscript according to order below:

First: write about the definition of endometrisis and challenges upon it's definition

Second: write about current various curative approaches in order to cure endometriosis and their weakpoints 

Third: write about the importance of the present manuscript based on previous information (in thr part Introduction) 

Note: authors in the part "introduction" should mention that why they have written this manuscript (the importance of this manuscript) 

 Response 5: The introduction was corrected on the bases of reviewer 1 comment

Point 6: in page 1, line 37-40 This part of introducrion disrupts the continuity of this part of the manuscript and does not have a proper connection with it's previous paragraph.Please rewrite or omit this part.

 Response 6: Authors decided to omit this part

Point 7: In page 2, line 57-60 Why authors have written this statistics in the part " 3. Different types of endometriosis"? It is better that this epidemiological data are written in their proper site of the manuscript.

Response 7: Each epidemiological data of peritoneal endometriosis, OMA and DIE was written in his proper site of the manuscript

Point 8: In page 3, line 124-127 Please add suitable reference at the end of the sentences of this part.

Response 8: Reference were added at the end of the sentences

Point 9: Page 3, line 128, 135 and 140 Please write these three sentences in the form of subheading like:

3.2.1.1.Superficial implants and invagination of the ovarian cortex 

3.2.1.2.Superficial implant and corpus luteum invasion 

3.2.1.3.The metaplasia of the invaginated mesothelial inclusions

Response 9: Sentences in this part were written in the form suggested by reviewer 1

Point 10: In page 4, line 145, part " 3.2.2. Diagnosis"

Please write the conclusion of this part based on scientific studies you have mentioned in this part.

Response 10: Conclusion were added to this part according to reviewer 1 suggestions

Point 11: Please add appropriate reference at the end of the sentences in page 4, line 156-159

Response 11: Appropriate reference was added at the end of this sentence  

Point 12: In page 4, line 171, there are two abbreviations called "AFC" and "AMH"

please expand these two abbreviation in brackets.

Response 12: The two abbreviation were expanded

Point 13: In page 4, line 194, part "3.3.1. Pathogenesis"

Please write a conclusion for this part based on previous information of mentioned part. (Please write what you conclude of the process of pathogenesis of DIE based on the data you have written in this part)

Response 13: Conclusion were added to this part according to reviewer 1 suggestions

Point 14: In page 5, line 219-221, line 230-235 and line 237-241

Please add appropriate reference at the end of the sentence(s).

Response 14: : Appropriates references were added at the end of these sentences

Point 15: In page 6, line 275-277 Please omit or tranfer this part to it's proper section of manuscript. 

In the part "Conclusion" authors should talk about the results of their work. The comparison of the results with other studies belongs to previous sections of manuscript.

Response 15 : The authors decided to not remove or transfer this sentence, because it’s needed to explain why the use of PET in the diagnosis of endometriosis is an interesting imaging tool as a future perspective. Conclusions were rewritten according to reviewer 1 suggestions.

Point 16: Please check references carefully (especially their titles and doi)

Response 16: All the references were carefully checked and doi were added

Reviewer 2 Report

The article presents an interesting topic that needs improvement-expansion.

1.The abstract requires an extension.
2. Materials and Methods, Search Strategy - The authors have to describe in detail what
databases were reviewed, what was the inclusion and exclusion criteria, how many
publications were found how many were qualified for the article. The authors should also
present this in the form of a figure.

3. The authors should add limitations

Author Response

Point 1: The abstract requires an extension.

Response 1: Abstract was extended. The first draft of the abstract was composed by 66 words, the revised version of the abstract is composed by 141 words

Point 2: “Materials and Methods, Search Strategy” - The authors have to describe in detail what databases were reviewed, what was the inclusion and exclusion criteria, how many publications were found how many were qualified for the article. The authors should also present this in the form of a figure.

Response 2: The “Screening of publications” paragraph was added to the article and Figure 1 (explaining seaching screening) was added to the article.

Point 3: The authors should add limitations

Response 2: The “Limitations” paragraph was added to the article.

Round 2

Reviewer 1 Report

Respected authors considered all of the comments and there is no more suggestions.